# Enhancing Wayfinding Performance in Existing Healthcare Facilities Using Virtual Reality Environments to Revise the Distribution of Way-Showing Devices

**Ammar Al-Sharaa** [1,*]**, Mastura Adam** [1]**, Amer Siddiq Amer Nordin** [2,3]**, Ameer Alhasan** [4]**, Riyadh Mundher** [5] **and Omar Zaid** [6]

1 Department of Architecture, Faculty of Built Environment, University of Malaya, Kuala Lumpur 50603, Malaysia; mastura@um.edu.my
2 Centre on Addiction Sciences (UMCAS), University of Malaya, Kuala Lumpur 50603, Malaysia; amersiddiq@um.edu.my
3 Department of Psychological Medicine, Faculty of Medicine, University of Malaya, Kuala Lumpur 50603, Malaysia
4 Faculty of Electrical and Electronic Engineering, University Tun Hussein Onn Malaysia, Batu Pahat 86400, Malaysia; ameer.nadhum91@gmail.com
5 Department of Landscape Architecture, Faculty of Design and Architecture, Universiti Putra Malaysia, Serdang 43400, Malaysia; arch.riyad@gmail.com
6 Department of Computer Science, Faculty of Computer Science and Information Technology, Universiti Putra Malaysia, Serdang 43400, Malaysia; omar.zoriqat@gmail.com
* Correspondence: ammoratawama@gmail.com

**Abstract:** Wayfinding is the process of navigating the environment by using the available environmental cues. The issue of wayfinding difficulty in large healthcare facilities has grabbed the attention of many researchers in terms of its potential taxing effects on both users and institutions alike. While the challenge of regulating the process of designing wayfinding systems is still an active field of investigation, there has been a paucity of research addressing the challenge of distributing way-showing items based on users' performance within healthcare environments. This study proposes a wayfinding enhancement scheme whereby users' wayfinding performance data forms the basis of the process of distributing way-showing items within an outpatient unit in Malaysia. Furthermore, two virtual reality experiments were carried out, the first representing the existing wayfinding system and the second representing the new distribution. A cross-comparison between the two sets of results was conducted to evaluate the effect resulting from altering the as-built wayfinding system. The results indicated an overall reduction of time consumed to reach the same destinations as well as lesser distances traveled within the environment resulting from the implementation of the new distribution. This study puts forward the concept of implementing virtual reality environments to address wayfinding systems' design challenges in healthcare facilities rather than relying on designers' intuition.

**Keywords:** hospital environment; outpatient unit; virtual reality; wayfinding

## 1. Introduction

Wayfinding can be defined as the process of finding one's way to a destination in a familiar or unfamiliar setting using any cues given by the environment [1]. Wayfinding has been an interest among researchers and professionals from different fields, namely architects, interior designers, cognitive psychologists, as well as researchers in the field of facilities management, to name a few. There were two types of wayfinding, identified as recreational and resolute [2]. Recreational wayfinding allows an individual the opportunity to solve problems that can be a source of satisfaction and enjoyment. Here, time is not an issue, while resolute wayfinding occurs when the main purpose is to find one's way

in the most efficient manner. Wayfinding in the real world could be broken down into a four-step process, namely: (1) orientation: when a person finds out where they are with respect to nearby landmarks and the required destination, (2) route selection: choosing a route that will eventually lead to the desired destination, (3) route control: the constant control and confirmation that the individual is following the selected route; and (4) recognition of destination: the individual's ability to realize that they have reached the desired destination [3].

Often when people are navigating in an unfamiliar environment, they have to find a destination without the help of an acquired mental map. They depend on external information, or what one calls knowledge in the world [4]. Such knowledge resides in the environment and is communicated through signs, guidance systems, and architectural clues. In many cases, people find it difficult to perform wayfinding tasks in an unfamiliar environment because they are not provided with adequate knowledge of the world. The main reason for environments being too complex to facilitate wayfinding is a deficiency of environmental clues [5]. They either lack sufficient wayfinding information, or way-showing items are poorly designed. Outpatient areas are the most visually cluttered areas in a hospital's interior environment and have the least exposure to the exterior environment [6].

Way-showing devices are an essential part of any wayfinding system, as they represent an interface whereby the institution can communicate with the users of the environment [7,8]. There are primarily three types of way-showing items, namely: directional devices that indicate the direction to a certain space or function, orientational devices that help the users to estimate their relative location within the environment, and defining devices that define spaces or locations [9]. Previous research has discussed the effects of various environmental elements on users' wayfinding, such as visual clutter [10,11], color [12], text [13], and verbal direction [14]. However, little research has focused on manipulating environments' wayfinding systems to achieve a better distribution of way-showing devices [15].

## 1.1. Virtual Environments in Architectural Research

Architectural visualization (AV) has an important role in design that is based on a long tradition of 3D computer-aided design modeling. Therefore, it has been expected that it will always extend into new types of 3D visualizations that will aid in the process of interacting with both existing and proposed environments [16]. The role of AV has been primarily as a presentation tool of complex information for the stakeholders of the design project. The objective of creating AV 3D models has been to create data-rich visualizations of a design project at its future stages. Moreover, AVs can provide a means of investigating the users' experience of the completed building at the early stages of the project to allow for early design alterations prior to the beginning of construction [17]. Recently, there has been an apparent effort by designers to integrate building information modeling (BIM) systems and values into their workflow to create data-rich 3D models with the aim of improving the understanding of design and planning data [18,19]. Utilizing these 3D data-rich models that have been modeled during the early design phases of the project into a virtual reality (VR) environment is then relatively a simple process after a potential format conversion and an integration within the virtual reality environment (VRE) platform, which makes using the final output at different phases of the design process possible [20]. The employment of CAD/BIM technologies in conducting post-occupancy evaluation has not been widely adopted by designers, building facilitators, and professionals in the fields of architecture, engineering, and construction (AEC).

A number of previous studies have used VREs to compare users' interaction with the real environment (RE) [21]. Further studies have simulated certain circumstances, such as emergency evacuation, to examine occupants' actions under these circumstances [22,23]. However, the main utilization of VR in architectural design and research has been so far as a design tool, particularly to facilitate cross-disciplinary decision making [24,25]. The main aim of employing VREs has been to harmonize different approaches and objectives of the

various potential stakeholders throughout the design process and offer an intuitive way to be exposed to the design data [26,27]. VR technology does not require specialist knowledge other than the know-how of integrating the 3D models into a control system. VREs have the potential to put forward a more inclusive approach to engaging various stakeholders with the design process of the built environment, offering the opportunity for a meaningful contribution by both users and institutions alike to shape the future of the architecture of the built environment.

### 1.2. Virtual Reality in Wayfinding Research

Wayfinding is an area of research that can potentially utilize emerging digital technologies such as VR technologies, especially when investigating the efficacy of wayfinding systems or assessing the difficulty of wayfinding in a certain setting or under a certain set of circumstances. VREs offer the potential to develop an examination technique to investigate users' interaction with wayfinding systems so that the knowledge can be applied in future wayfinding design initiatives and enhance wayfinding performances of existing buildings. A large amount of the previously published research that has employed VR technology was focused on employing VRE experiments as a simulation of wayfinding tasks in REs. However, in some cases, wayfinding tasks were not set up to represent REs, where the participant's performance was observed while navigating through a maze-like environment [28].

Furthermore, recent research initiatives have emerged attempting to investigate the performances of modified environments by controlling a certain number of limited variables, such as signage or lighting [29,30]. Wayfinding research in the built environment is still an active area of research and remains to represent a practical problem for the users of large or complex buildings, such as museums, libraries, airport terminals, shopping centers, and hospitals [31] or when users are subject to a set of stressful circumstances such as emergency evacuation [30,32]. In such circumstances, the response of occupants to the wayfinding task at hand is largely dependent on occupants' engagement with the built environment, including the multi-sensory cues to orientation and navigation provided by the environment.

Other published studies [33,34] have employed a 3D model displayed on a monitor or a screen, in which the participants either passively followed a predetermined route or attempted to complete a particular navigation task. However, some studies have attempted to compare users, behavior, or performance in a real environment [35] with a VR version of the environment. Furthermore, there was a paucity of wayfinding research that has employed immersive navigable VREs. It is worth noting that most of the researchers who used immersive VREs as a research tool were not explicitly focused on indoor navigation performance [36]. Previously published research suggests that there is an agreement that VREs provide an experimental analog that simulates the as-built or as-planned real environments (REs) [37].

### 1.3. The Internal Referral System within Outpatient Units in Malaysia

A set of documents was published by the Malaysian Ministry of Health (MoH) [38] to form a guideline on which healthcare facilities' information management systems can be based. The documents are not focused on wayfinding in healthcare facilities as much as way information is communicated amongst the stakeholders within the hospital. However, the documents highlighted a set of recommended institutional practices regarding the users' procedures to go through when visiting a healthcare facility from registration until billing; see Figure 1.

This study proposes a way of showing an items distribution scheme to be implemented in healthcare facilities of Malaysia, with outpatient units being the subject of experimentation and the main focus, whereby a distribution of way-showing items is based on the outcomes of a virtual reality experiment to evaluate the current wayfinding system. The

next section of this paper highlights the methods incorporated in this study, including selecting the case study, data collection techniques, and data analysis techniques.

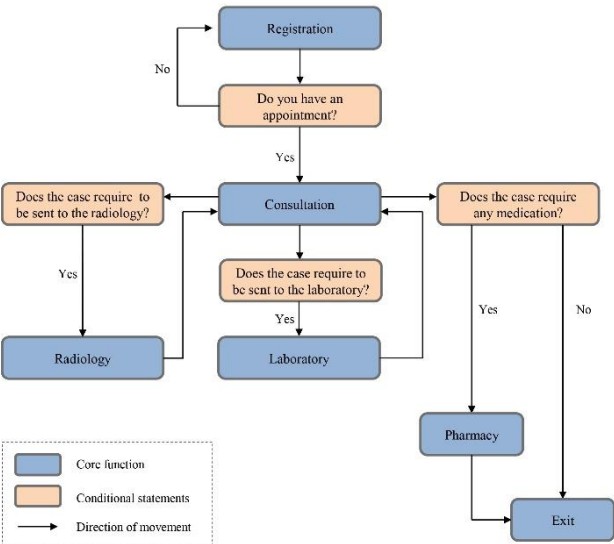

**Figure 1.** Outpatient units' high-level workflow based on the MoH's information management system guidelines [39] illustrates the high-level procedure of handling the cases from an institutional perspective to inpatient and outpatient care in hospitals. PMS focuses on the document's content pertaining to outpatient cases treatment and referral procedures.

## 2. Materials and Methods

This study investigates the viability of considering visibility as the basis of how the placement and orientation of wayfinding assistance devices can take place. A case study was selected for the purpose of implementing the proposed distribution technique. The upcoming section highlights the case study characteristics, the data collection techniques, and the data analysis techniques.

### 2.1. Ruka Building of University Malaya Medical Centre (UMMC) as a Case Study

The case study selected in this research is RUKA. RUKA is a component of an outpatient unit of UMMC also known as Pusat Perubatan Univrersiti Malaya (PPUM), which is a government teaching hospital in Malaysia. The hospital was established in 1987 and provides medical care for the Klang Valley area.

Since starting its operation, the facility has attracted a growing number of patients annually; however, no major renovation has been carried out. The clinic is divided into two floors—the totals on the ground and first floor are 1034 m$^2$ and 598 m$^2$, respectively. Essentially, there are three main zonings of interior space that serve the patients—the reception lobby, the consultation area, and the pharmacy area. Each of these areas has seating provisions for waiting. Ground floor is where the drop off, registration counter, consultation area, and pharmacy are located.

The first floor layout is a repeat of the consultation area on the ground floor except that it only has 15 designated consultation rooms. The pharmacy area is located on the right side of the registration lobby. It has a waiting area of 108 m$^2$, located next to the exit door. This door also leads patients to the extension of the pharmacy waiting area located outside of the area. This area is an extension built out of the original design, which used to function as a corridor space leading to the specialist clinics and ward area of the general hospital.

Typically, a patient seeking for treatment will go through the process of registration, waiting prior to consultation with the doctor, and a second wait to collect the medicine. This area is divided into five areas: drop off, registration counter, waiting area, consultation rooms, laboratory, radiology, and pharmacy.

*2.2. Data Collection Techniques*

Three data collection techniques and two data analysis techniques were incorporated in this study. The first data collection technique took place in the form of collecting the architectural drawing from the engineering department of UMMC. The second data collection technique was in the form of surveying the interior of the building in order to locate the current location and orientation of the way-showing items in the existing outpatient unit's building; see Figure 2. The third data collection technique was in the form of a navigable virtual environment representing the built environment in both states being the existing distribution as well as the visibility based proposed distribution. The virtual environments were created based on the previously collected architectural drawings, whereby three-dimensional (3D) models were created using a modelling software called 3ds Max. The 3D models were then exported to an open-source game development platform to create an on-demand navigable virtual environment whereby a certain wayfinding tasks can be given to the subjects by the investigator. The virtual environment then is capable of calculating distances traveled by subjects, the time taken to reach their destination, the number of pauses taken during their journey, and the number of map calls.

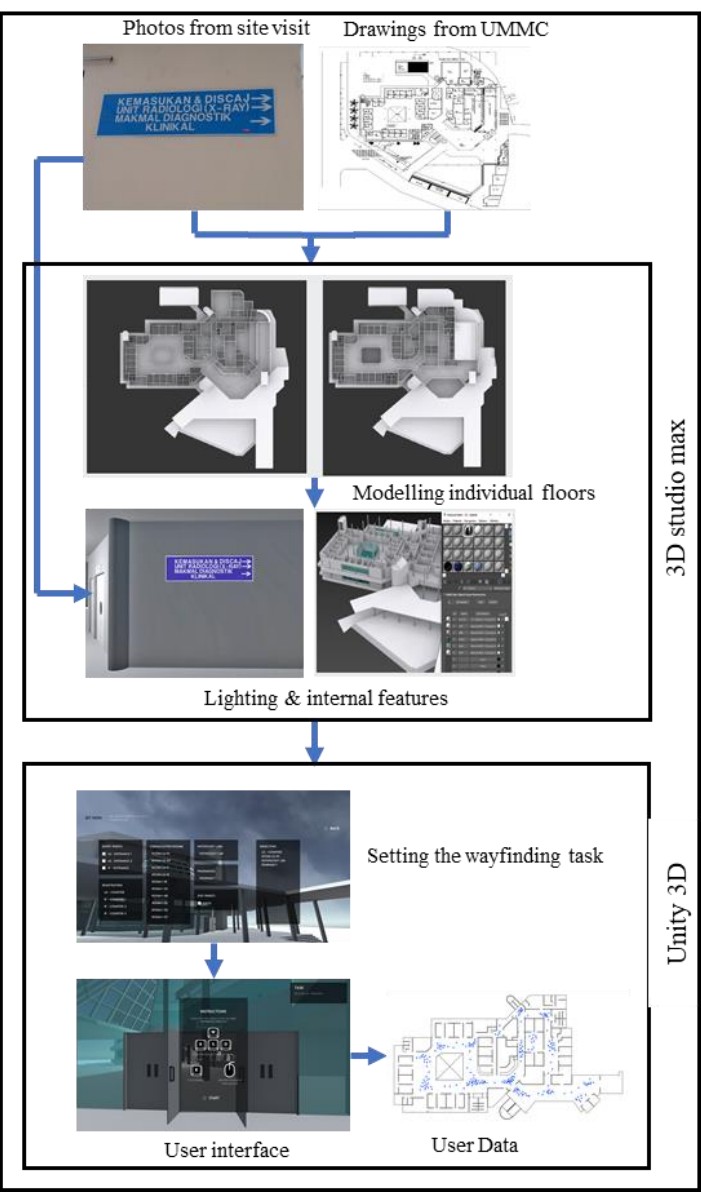

**Figure 2.** The setup of the Ruka 3D model in order to export to unity 3D.

The lighting, the camera/user point of view (PoV), and the user controls were all set within the Unity environment. The navigable areas were defined using a command function known as mesh baking in which the extent to which the user can move is predefined. The same is carried out in the process of creating the second iteration of the virtual walkthrough containing the developed wayfinding system; see Figure 2.

2.2.1. Constructing the Virtual Model

Wayfinding tasks in a virtual environment are aimed to test and evaluate the effectiveness of the proposed set of guidelines to be incorporated when developing a wayfinding system for outpatient units. However, here, all of this is happening in a certain setting, which is represented by the Malaysian public healthcare system. The institutional guidelines represented by the Ministry of Health's information management system guidelines gives an indication of the expected workflow within each department of the hospital outpatient units included; see Figure 1, Table 1.

**Table 1.** Description of the paths selected for the VRE.

| No. | Entrance Level | Consultation Room | Laboratory | Pharmacy | Minimum Number of Changes in Elevation |
|-----|----------------|-------------------|------------|----------|----------------------------------------|
| 1 | Ground | 1 | Yes | Yes | 0 |
| 2 | Ground | 5 | Yes | Yes | 0 |
| 3 | Ground | 11 | Yes | Yes | 0 |
| 4 | Ground | 15 | Yes | Yes | 0 |
| 5 | 1st | 18 | Yes | Yes | 1 |
| 6 | 1st | 23 | Yes | Yes | 1 |
| 7 | 1st | 28 | Yes | Yes | 1 |
| 8 | 1st | 32 | Yes | Yes | 1 |
| 9 | 1st | 35 | Yes | Yes | 1 |
| 10 | 1st | 37 | Yes | Yes | 1 |
| 11 | Ground | 18 | Yes | Yes | 2 |
| 12 | Ground | 25 | Yes | Yes | 2 |
| 13 | Ground | 28 | Yes | Yes | 2 |

This internal high-level workflow was embedded in the virtual wayfinding experiment. In the virtual environment, the principal investigator represented by the researcher can assign a certain wayfinding task that is still based on the high-level workflow, yet unique to each user to explore the effectiveness of an empathetic internal wayfinding system that considers users' nuanced characteristics. The virtual environment, while it cannot gauge peoples' physical mobility, can provide a better understanding of how a referral system that considers different demographic factors can aide in the process of wayfinding. The principal investigator controls the assigned wayfinding tasks using a simple user interface developed for the purpose. The virtual wayfinding tasks are comprised of a number of minor navigation tasks in line with the referral system deduced from the MoH's information system management guidelines in outpatient units demonstrated in Figure 3.

2.2.2. Translating the Internal Referral System into A Wayfinding Virtual Walkthrough Experiment

Based on the MoH's high level workflow, a set of wayfinding tasks within the environment was tested in which a set of typical wayfinding tasks were represented. These wayfinding tasks can be summarised in Table 1, whereby generic wayfinding tasks within RUKA's environment were tested.

Moreover, Figure 3 illustrates the user interface used by the investigator to select paths for the subjects.

Furthermore, Figure 4 illustrates the selected zones and functions in the virtual experiment, which are are highlighted.

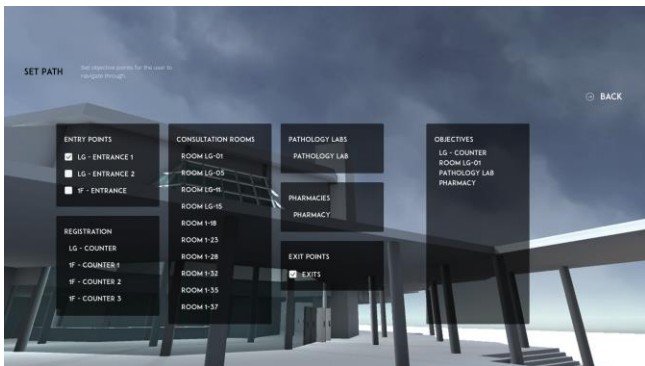

**Figure 3.** The wayfinding experiment's user interface used by the investigator to set up the paths.

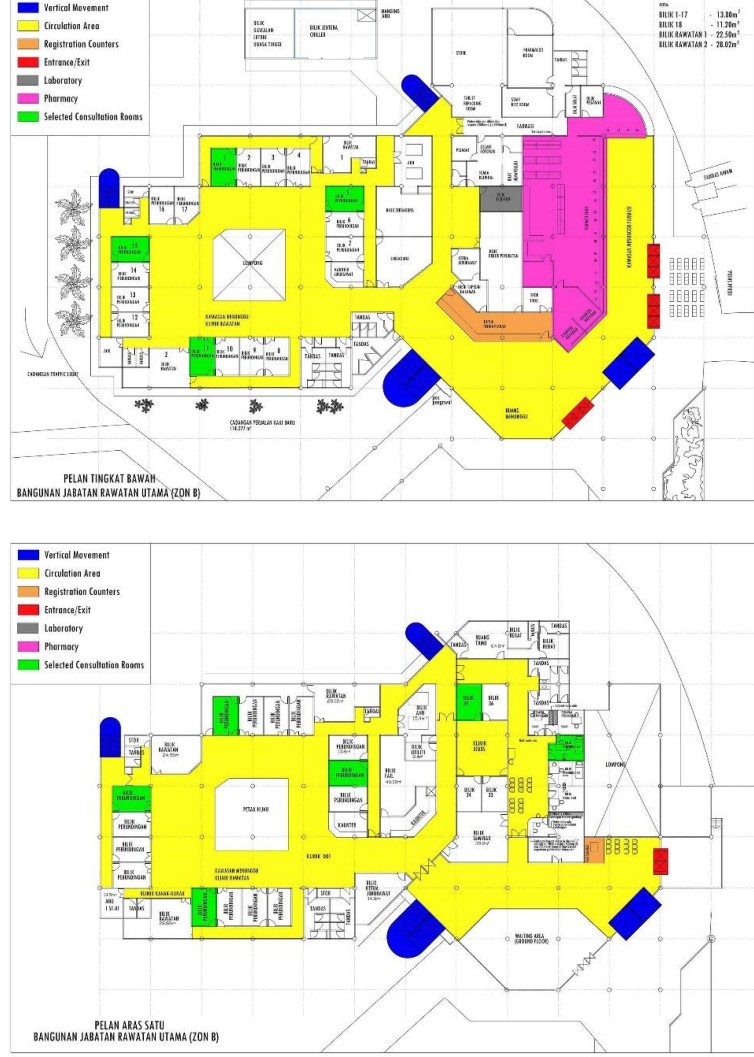

**Figure 4.** The selected zones and functions in the virtual experiment are highlighted.

### 2.2.3. Virtual Walkthrough Experiment Protocol

The protocol employed in the wayfinding virtual experiment can be deduced from the MoH's information management system development guidelines; a typical wayfinding task was translated into the workflow that subjects were required to follow in the virtual experiment. The first step in the protocol was required from the principal investigator to pre-set the wayfinding journey for the subjects. The first setting was selecting the entrance. Then the investigator needs to set the registration counter as the next wayfinding

sub-task within the environment, with the registration being an integral process in the overall workflow. After stopping by the registration counter, patients will be referred to a consultation room, where a consultant will provide the appropriate required procedure. The final stage is to direct the subject to leave the building. In this study's typical outpatient movement, no external referrals were integrated in the process. Therefore, patients will be proceeding to the laboratory area. While it is not an absolute requirement for all patients to require a visit to the laboratory, it is seen as a highly important space within the outpatient unit workflow. See Figure 5.

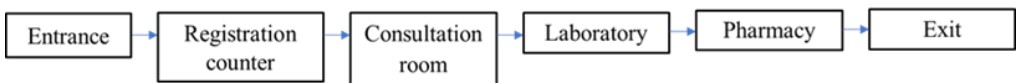

**Figure 5.** Exemplary wayfinding task flow within the VRE.

The following task set by the investigator and ought to be undertaken by subjects is the process of visiting the pharmacy counter for the purposes of collecting medication, if needed. Lastly, subjects are required to perform an exit from the hospital's virtual premises. A set of collected data will be gathered automatically by the VWE application itself, gathering information on subjects', time, distance, a trace of the selected path by the subject (see Figure 6), a record of points where the users have used the map option, and number of times and locations where users have asked for directions. The time and distance of each subject is saved into an external SAV extension file.

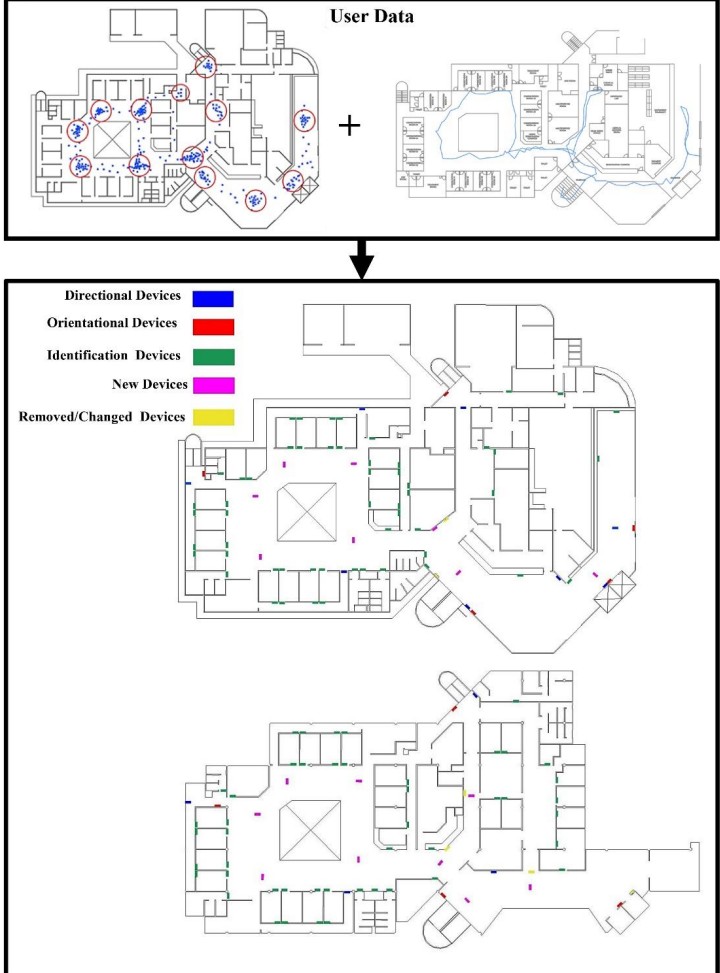

**Figure 6.** The distribution and type of way-showing devices and an example of the user data its based upon.

While subjects' selected paths are registered through real-time tracking of users' navigation routes, users' pausing points are registered on the floors' layouts. Pausing points indicate areas of decision making or confusion. The same procedure then applied for the second set of users going through the virtual experiment with wayfinding items located at the spaces where decision making points are located at the first VRE. Both sets of data were then analysed and compared to indicate the effectiveness of the revised wayfinding system in which the newly developed signage system is based upon. See Figure 6.

2.2.4. Virtual Experiment User Control, Instruction, and Information Display

The user controls were set to be a keyboard and a mouse-based control due to its popularity amongst computer users, especially in the users' control of video games where the user uses the (W, A, S, D) keys for movement on the two-dimensional plane, which is represented by the baked mesh or the navigable space. However, the direction of vision and movement is controlled by the movement of the mouse. This information was displayed to the subjects as a user instruction, whereby he/she can read the instructions and then click to proceed; see Figure 7.

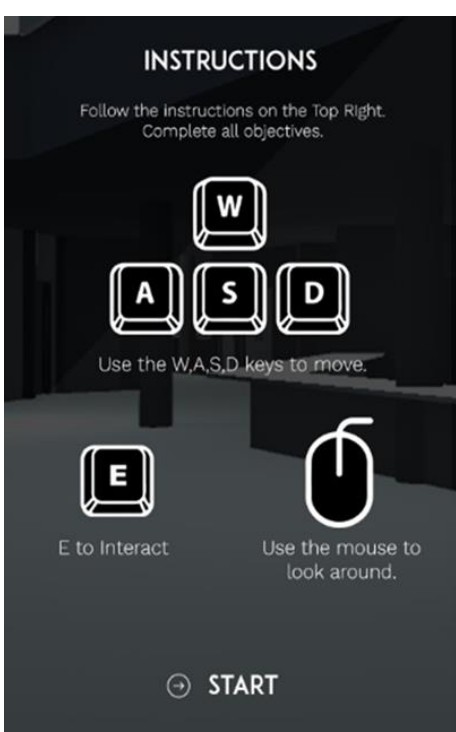

**Figure 7.** The user instructions display prior to starting the wayfinding task.

Moreover, the instructions were displayed in the form of a pop-up card in the left upper corner of the screen; see Figure 8.

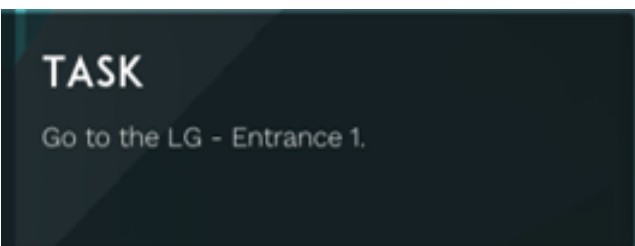

**Figure 8.** Illustration of the task commands required from the subject to perform in the form of a pop-up card.

In the pop-up cards, instructions for the subjects are written, instructing the subject to go to a certain place that correspond to the function mandated by the workflow highlighted in Figure 1.

These pop-up cards confirm the wayfinding task achievement when the subject reaches his/her destination. Furthermore, the subjects were instructed to press the M button in order to display a map showing the subjects' respective locations for orientation purposes in case they lost their orientation; see Figure 9.

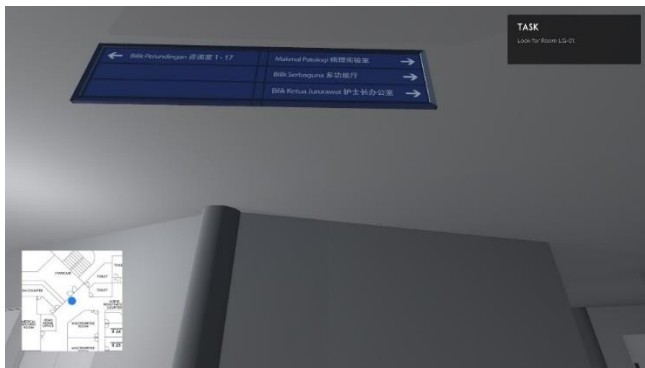

**Figure 9.** The map-call function display on the bottom left side of the screen.

### 2.3. Data Analysis Techniques

The data analysis techniques incorporated in this study were mainly statistical, whereby the statistical data analyses of two sets of users were collected representing distances covered by users in both virtual environments as well as the time taken to reach each destination. The statistical analysis was performed using IBM SPSS version 25, whereby a set of statistical techniques was employed to investigate whether there was an improvement in the distances and/or the time taken to reach the destination in the task.

The first set of techniques were the skewness and kurtosis to investigate the normality of distribution, while the second test was a *t*-test. Furthermore, an eta squared values investigation was carried out to investigate the size of the impact.

### 3. Results

The results of this study can be divided into three sections, namely the results of the first VRE, the results of the second VRE, and a cross comparison between the two VREs.

### 3.1. Results of the First VRE of Ruka Outpatient Area

The first VRE was conducted for the existing environment of the RuKa building which is the first virtual experiment to be conducted in this study. A total of 260 participants participated in the VRE; 130 of them experienced the building with its current wayfinding system, while the other half experienced the building with a developed wayfinding system based on the response of the first group. Ten observations were made per each of the 13 selected paths. The first virtual walkthrough experiment was conducted for the existing environment of the RuKa building, which is the first outpatient area to be included representing the first VRE.

First VRE Descriptive Statistics

The VRE distance results registered a minimum average of 214.7 m in task (1) with a standard deviation (stdv.) value of 12.19 and a maximum of 417.5 m in task (13) with a stdv. value of 46.46. Furthermore, the lowest average time registered was in task (2), registering 234.8 m and a stdv. value of 29.2 and a maximum of 480.1 m in task (11) with a stdv. of 61.87. See Table 2.

**Table 2.** Summary of the data recorded from the Ruka 1st VRE.

| Task No. | Average Distance (Meter) | Stdv. | Average Time (Seconds) | Stdv. | NoP | NoMC |
|---|---|---|---|---|---|---|
| 1 | 214.7 | 12.19 | 249.1 | 15.68 | 38 | 13 |
| 2 | 233.9 | 15.34 | 234.8 | 29.20 | 37 | 15 |
| 3 | 226.6 | 13.53 | 254.8 | 31.03 | 35 | 13 |
| 4 | 237.5 | 4.60 | 282.1 | 36.30 | 39 | 11 |
| 5 | 324.4 | 64.95 | 350.9 | 49.14 | 43 | 14 |
| 6 | 330.2 | 35.40 | 378.4 | 53.74 | 47 | 21 |
| 7 | 288.3 | 27.62 | 341.6 | 42.52 | 41 | 15 |
| 8 | 339 | 36.38 | 360.9 | 52.75 | 46 | 19 |
| 9 | 355 | 47.72 | 371.7 | 40.08 | 43 | 20 |
| 10 | 368.8 | 48.78 | 448.4 | 83.34 | 49 | 22 |
| 11 | 398.5 | 40.57 | 480.1 | 61.87 | 48 | 17 |
| 12 | 410.1 | 55.83 | 434 | 40.64 | 55 | 21 |
| 13 | 417.5 | 46.46 | 436.2 | 30.19 | 53 | 22 |

Furthermore, the VRE have registered a total of 574 pauses by the respondents, registering a minimum of 35 pauses in task (3) and a maximum of 55 pauses in task (12). The respondents averaged 4.41 pauses per task in all the tasks given. Moreover, the VRE registered a total of 233 instances where the users used the pop-up orientation map option, averaging a 1.79 map call up per task across all tasks. The minimum no. of map requests per task was registered in task (4), and the maximum was registered in task (13). See Table 2.

The pauses can be mapped on the building layout and to investigate the patterns of the pauses; Figure 10 illustrates the location of the pauses mapped on the building layout as well as highlights areas where the subjects made more pauses than others.

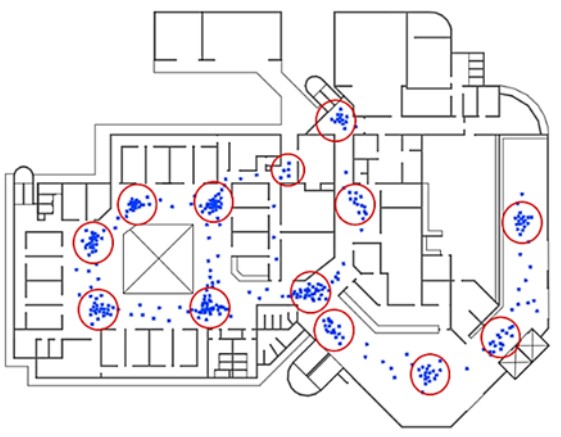

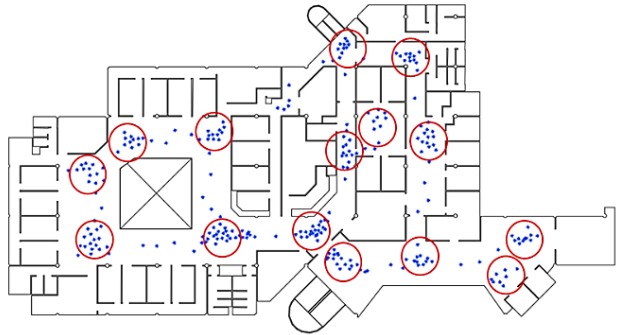

**Figure 10.** Illustration of the location of the pauses mapped on the building layout of the 1st VRE.

It is worth mentioning that more pauses were registered in the ground floor than the first floor due to the fact that all the users had to exit and visit certain areas such as the laboratory and the pharmacy.

The locations where the subjects opted for calling an orientation map of maps requirements were also mapped on the building layout, indication areas where the subjects struggled with their orientation with respect to both their starting points or their destinations; see Figure 11.

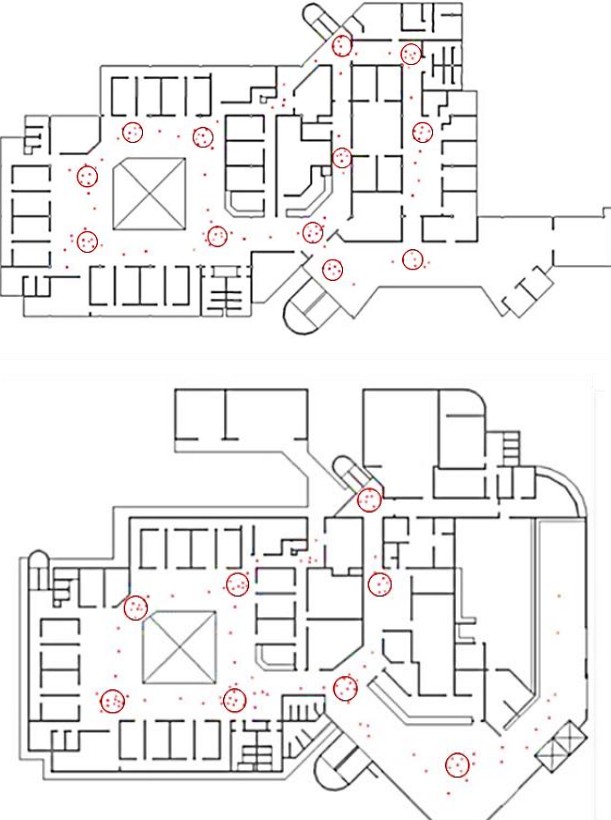

**Figure 11.** Illustration of the location of the map calls mapped on the building layout of the 1st VRE.

These areas then are identified as areas where users required external help due to the lack of information available. Several changes were made to the existing environment whereby an orientational device was added at the location where subjects lost their orientation, provided the space did not contain an orientation device. In cases where an orientation device existed, and subjects still lost their orientation, the same device was kept with a minor change of the device's orientation to increase the device's visibility from the location where the pauses were concentrated.

As for places where users made a large number of pauses, directional devices were added to highlight the direction where main functions exist, such as the pharmacy, the laboratory, and the consultation rooms. In cases where a directional device existed, a minor change to the device's orientation took place to face the location to increase visibility from the location where the largest number of pauses took place.

### 3.2. Results of the second VRE of Ruka Outpatient Area

The second VRE was conducted for the RuKa building, which is the first outpatient area to be included representing the first case study. A number of adjustments took place as to the location of way-showing items in the building based on the data collected from the first VRE.

Second VRE Descriptive Statistics

The VRE distance results registered a minimum average of 209.9 m in task (1) with a stdv. value of 13.11 and a maximum of 331.6 m in task (13) with a stdv. value of 15.02. Furthermore, the lowest average time registered was in task (1) registering 220.6 m and a stdv. value of 21.63 and a maximum of 352.2 m in task (13) with a stdv. of 17.63.

Furthermore, the VRE registered a total of 488 pauses by the respondents, registering a minimum of 33 pauses in task (1) and a maximum of 43 pauses in tasks (12) and 1–18. The respondents averaged 3.75 pauses per task in all the tasks given. Moreover, the VRE registered a total of 141 instances where the users used the pop-up orientation map option, averaging a 1.7 map call up per task across all tasks. The minimum no. of map requests per task was registered in task (2), and the maximum was registered in task (13). See Table 3.

**Table 3.** Summary of the data recorded from the Ruka 2nd VR.

| Task No. | Average Distance (Meter) | Stdv. | Average Time (Meter) | Stdv. | NoP | NoMC.1 |
|---|---|---|---|---|---|---|
| 1 | 209.9 | 13.11 | 220.6 | 21.63 | 33 | 7 |
| 2 | 228.6 | 19.97 | 222 | 13.05 | 35 | 2 |
| 3 | 223.5 | 14.47 | 256 | 32.03 | 39 | 6 |
| 4 | 231.2 | 10.33 | 266.5 | 15.96 | 39 | 6 |
| 5 | 242 | 44.41 | 263.6 | 63.06 | 43 | 8 |
| 6 | 226.2 | 16.00 | 224.6 | 15.36 | 37 | 5 |
| 7 | 221.1 | 19.39 | 247 | 36.76 | 34 | 6 |
| 8 | 230.5 | 12.88 | 256.9 | 16.56 | 36 | 7 |
| 9 | 253.2 | 33.49 | 292.7 | 34.68 | 37 | 17 |
| 10 | 312.4 | 21.31 | 338.3 | 21.82 | 39 | 18 |
| 11 | 313.1 | 19.68 | 351.9 | 29.92 | 35 | 19 |
| 12 | 294.9 | 16.36 | 312.3 | 13.91 | 43 | 17 |
| 13 | 331.6 | 15.02 | 352.2 | 17.63 | 38 | 23 |

The pauses can be mapped on the building layout and investigate patterns of pauses. Figure 12 illustrates the location of the pauses mapped on the building layout and highlights areas where the subjects made more pauses than others.

It is worth mentioning that more pauses were registered in ground floor than the first floor due to the fact that all the users had to exit and visit certain areas such as the laboratory and the pharmacy. The locations where the subjects opted for calling an orientation maps of maps requirements were also mapped on the building layout; for indicated areas where the subjects struggled with their orientation with respect to their starting points and/or their destinations, see Figure 13.

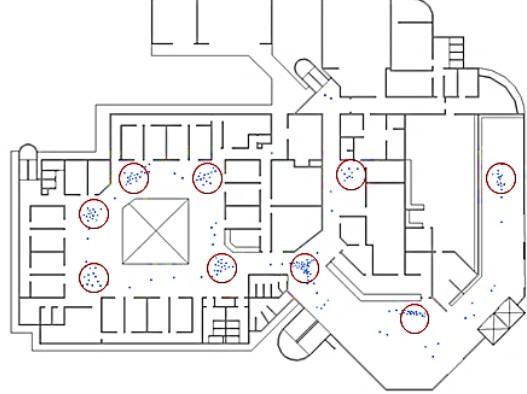

**Figure 12.** *Cont*.

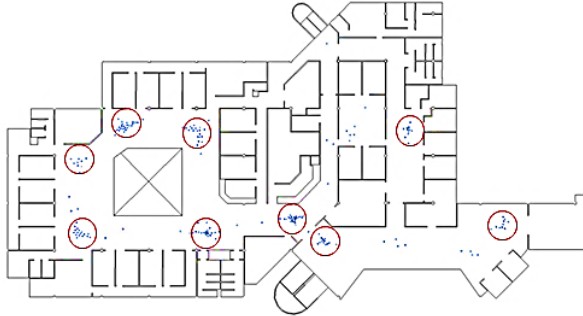

**Figure 12.** Illustration of the location of the pauses mapped on the building layout of the 2nd VRE.

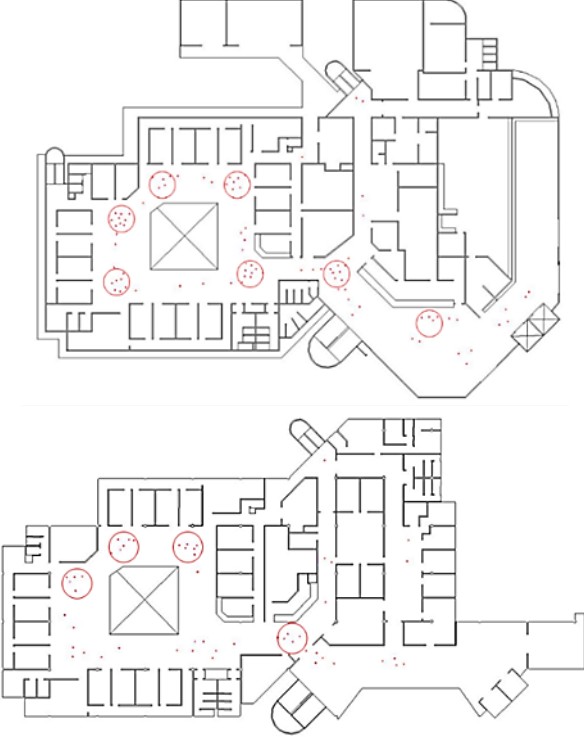

**Figure 13.** Illustration of the location of the map calls mapped on the building layout of the 2nd VRE.

*3.3. Cross Comparison between the 1st and the 2nd Ruka VREs*

Comparing the two sets of data collected via the two VREs by conducting an independent sample *t*-test will indicate whether a statistically significant change has occurred in the overall performances of participants by introducing changes on the wayfinding system at Ruka. Furthermore, the magnitude of this effect size can be measured by the eta squared formula. Investigating Levene's Test for Equality of Variances indicates Sig. values below 0.05, the average distance, NoP, and the number of map calls. This indicates that the hypothesis of the variances being equal cannot be rejected. Moreover, the *t*-test for equality of means indicates that the Sig. (2-tailed) values indicates that the null hypothesis of the means being equal cannot be accepted. Conversely, the Sig. value for the average time taken to complete the task registered a value of 0.097; therefore, the hypothesis of the variances being equal cannot be rejected. Furthermore, the *t*-test Sig. (2-tailed) value indicates that the null hypothesis of the means being equal cannot be accepted registering a value of 0.007. See Table 4.

**Table 4.** *t*-test results for the two VREs.

| Independent Samples Test | | Levene's Test for Equality of Variances | | *t*-test for Equality of Means | | |
|---|---|---|---|---|---|---|
| | | **F** | **Sig.** | **t** | **Sig. (2-Tailed)** | **Mean Difference** |
| Average Distance (meter) | Equal variances assumed | 4.819 | 0.038 | 2.738 | 0.011 | 63.561 |
| | Equal variances not assumed | | | 2.738 | 0.013 | 63.561 |
| Average Time (seconds) | Equal variances assumed | 2.983 | 0.097 | 2.994 | 0.006 | 78.338 |
| | Equal variances not assumed | | | 2.994 | 0.007 | 78.338 |
| NoP | Equal variances assumed | 6.923 | 0.015 | 3.452 | 0.002 | 6.615 |
| | Equal variances not assumed | | | 3.452 | 0.003 | 6.615 |
| NoMC. | Equal variances assumed | 11.031 | 0.003 | 2.893 | 0.008 | 6.308 |
| | Equal variances not assumed | | | 2.893 | 0.009 | 6.308 |

Calculating the partial eta squared values indicates the effect size of the independent variables to the effect size of the dependent variable. Generally speaking, an eta squared value below 0.13 is small, 0.13–0.26 is medium, and 0.26 and above is considered a high effect. See Table 5.

**Table 5.** Partial eta squared values for the size of the effect.

| No. | Item | Eta Squared | Effect Size % |
|---|---|---|---|
| 1 | Distance (meters) | 0.2381 | 23.81 |
| 2 | Time (seconds) | 0.2724 | 27.24 |
| 3 | NoP | 0.3317 | 33.17 |
| 4 | Number of map calls | 0.259 | 25.9 |

The values indicate that eta strongly affects the time measurement and the number of pauses, registering eta values of 0.2724 and 0.3317, respectively. However, a medium effect was observed on the distances covered and the number of map calls.

## 4. Discussion

The cross comparison carried out between the VREs has shown a statistically significant difference in performance across all measurements, whereby the Levene's test for the equality of variances has shown Sig. values below 0.05, indicating that the variances cannot be assumed equal except for the average time (seconds) measurement (Sig. = 0.097), while the *t*-test for the equality of means indicated Sig. (2-tailed) values bellow 0.05 in all measurements, indicating that the assumption of the equality of mean values, which represents subjects' performances, cannot be assumed equal; see Table 5. The positive statistically significant mean difference registered indicates that the effect of the intervention made on the wayfinding system in the Ruka building had a significant effect on subjects' performances, registering an average reduction in distance (meters) of >63.56 s, an average time reduction of >78.33, an average reduction in NoP > 6.61, and an average reduction in no. of map calls > 6.30. The analysis indicates that data collected from the users via the first and second VREs in upgrading the wayfinding system for the Ruka outpatient unit has had a significant outcome. Therefore, the employment of the VRE approach is to be recommended for future studies and design initiatives taking on the subject matter.

The study's results are consistent with the findings of [40], with regards to the effectiveness of using VREs to investigate wayfinding systems in existing buildings. Furthermore, the results of the study agree with [36], with regards to considering pauses as a sign for cognitive load amongst the study's subjects, whereby the second VRE had a lower number of pauses on average, and users were more efficient in their wayfinding, which can be seen by finishing their tasks during less time on average. Moreover, the study's results

are consistent with [41], regarding the importance of orientation devices in the process of users' re-orientation.

## 5. The Study's Limitations

The results of this study contribute to the area of development of wayfinding systems design and investigation procedures at healthcare facilities. However, the findings of this study are not without limitations. Firstly, the findings are limited by the sample size; a larger sample size could produce a more generalizable set of findings. Secondly, the findings of this study are limited by the fact that we targeted the users of a single public outpatient unit (RUKA). Further studies in which the same procedure can take place on a different environment could help with to generalize the procedure as a procedure that can fit all outpatient units within the Malaysian setting. Thirdly, this study employed a VRE approach to gauge the performance and detect the areas where improvement can take place on the wayfinding system of a Malaysian public outpatient unit. Virtual reality experiments are not without their limitations; while they serve as a good simulation tool, previous studies have suggested that the issue of visual fidelity has always been a concern. While in this study we made sure that our VR environment was as close to the real environment as possible, there is still a limited computing power available. Future research initiatives with more computing power than what this study has employed could produce a more accurate outcome. This study measured the performance of patients by incorporating only four measurands, namely: time, distance, the number of pauses, and the number of map calls. Further studies are encouraged to investigate the above mentioned way-showing items' properties alongside other properties such as the content of way-showing devices. Moreover, other wayfinding performance indicators have not been tested, such as the number of detours, the hesitation time, and error rates; future research initiatives considering those measurands may provide a more accurate assessment of wayfinding performance. Finally, this study only included patients of and visitors to a Malaysian public hospital. This led us to neglect employees' performance within the environment, which may be different from those of patients and visitors due to employees' familiarity with the environment.

## 6. Conclusions

This study aimed to enhance the efficacy of a wayfinding system in the Ruka outpatient unit of the University of Malaya Medical Center (UMMC). The study employed a virtual environments experimental protocol based upon the Malaysian healthcare referral system to measure the performance of users. The approach of this study could be employed as a guide to develop a wayfinding system in public hospitals that shares a similar referral scheme to the case study employed in this study. Furthermore, the results have shown the effectiveness of employing VREs as a post-occupancy evaluation (POE) technique to evaluate the efficacy of way-showing systems in outpatient units of public hospitals. The statistical analysis employed to analyze the collected VRE data indicates a significant improvement in the metrics selected by the study. The mean results indicate an improvement in the time consumed to finish the wayfinding tasks and the distances covered by the subjects to finish the task, while the subjects made less pauses and required less external assistance. Furthermore, the results of this study illustrate the importance of information display as an essential part of wayfinding systems in general, with way-showing items' availability, location, and orientation being the focus of this study. The results of this study also highlights that a constant evaluation of wayfinding systems is necessary to keep the wayfinding system up to date with the environmental changes in the hospital's physical environment.

**Author Contributions:** Conceptualization, A.A.-S., M.A. and A.S.A.N.; methodology, A.A.-S., M.A. and A.S.A.N.; software, A.A.-S., A.A., O.Z. and R.M.; formal analysis, A.A.-S.; writing—original draft preparation, A.A.-S.; writing—review and editing, A.A., O.Z. and A.A.-S.; supervision, M.A. All authors have read and agreed to the published version of the manuscript.

**Funding:** This research received no external funding.

**Institutional Review Board Statement:** Ethical approval was obtained through university of Malaya medical research ethics committee, MRECID NO: 20191017-7926.

**Informed Consent Statement:** Informed consent was obtained from all subjects involved in the study.

**Data Availability Statement:** Not applicable.

**Conflicts of Interest:** The authors declare no conflict of interest.

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
