# Peer review of "Enhancing Wayfinding Performance in Existing Healthcare Facilities Using Virtual Reality Environments to Revise the Distribution of Way-Showing Devices"

_buildings, doi:10.3390/buildings12060790_

Round 1

Reviewer 1 Report

Dear authors thank you for an interesting article, which describes an innovative approach to optimizing wayfinding performance in the existing healthcare facility. The proposed method incorporating Virtual Reality technology is promising, however, a vast number of possible scenarios (in a single building, or hospital department), can engage a relatively significant amount of work to receive good results.  

As the authors mentioned for those particular examples a number of 260 people took part in the tests - It is worth noting that such a significant amount of people engaged in the test proves the authors' vision and professionalism.

Research designed by the authors confirms their thesis, that virtual reality could be a tool for improving wayfinding performance in the existing healthcare facility.

Below are a few suggestions regarding the structure of the text: 

- lines from 223 to 239 could be skipped - this part of the text presents obvious information not so important in this article, then the points 2.2 and 2.2.1 can be combined,

- lines 287 to 310 - could be more attractive for the readers if will be depicted as a block scheme - consider a change,

- lines from 174 to 196 - could be skipped. Reducing this part will help to reduce in general the article, which is quite long,

- figures 9, 10, and 11 have to be improved - please increase the quality to achieve the same result as in figure 8.

To accept an article for publication, I am asking the authors to answer the following questions:

- if I good understand the narration your method of arranging the environment and guidance system is better than the official rules treated as a standard in designing of healthcare facility - please comment on it,

- in your research, you used a classic approach to the VR, well known for a classic game's interface (keyboard and mouse), maybe using VR goggles could generated a more realistic experience - please comment on your decision,

- to bring the process closer to a larger scale implementation, it is necessary to introduce some automatisms (perhaps machine learning based on data collected from real users, or programmable software agents) - are you planning any further development to make this method faster and more effective?

Author Response

Reply to reviewers’ comments

First and foremost, we would like to thank the reviewer for his valuable comments. 

Reviewer no.1

  • Comment: lines from 223 to 239 could be skipped - this part of the text presents obvious information not so important in this article, then the points 2.2 and 2.2.1 can be combined,

Reply: the text has been omitted from the article. As for sections 2.2 and 2.2.1, they have been combined at the request of the reviewer.

  • Comment: lines 287 to 310 - could be more attractive for the readers if will be depicted as a block scheme - consider a change,

Reply: a figure was added to the text illustrating the process in question

  • Comment: - lines from 174 to 196 - could be skipped. Reducing this part will help to reduce in general the article, which is quite long,

Reply: the text has been omitted from the article

  • Comment: - figures 9, 10, and 11 have to be improved - please increase the quality to achieve the same result as in figure 8.

Reply: the graphs have been processed to increase their image sharpness, hopefully it is satisfactory.

  • Comment: if I good understand the narration your method of arranging the environment and guidance system is better than the official rules treated as a standard in designing of healthcare facility - please comment on it,

Reply: your understanding of the method is on point; however, we have to mention that there are no specific guidelines ruling the process of designing or distributing way showing devices in Malaysia.

  • Comment: in your research, you used a classic approach to the VR, well known for a classic game's interface (keyboard and mouse), maybe using VR goggles could generated a more realistic experience - please comment on your decision,

Reply: there are both pros and cons to using each interaction interface, the googles are considered as just a display method, the difference would have been the usage of sticks to control the motion, which for accurate motion registration we would have needed to scatter VR reflectors within the place. The data collection for this study took place during the middle of the last year, where collecting data inside the hospital premises was prohibited due to the safety measures hospitals took to limit the spread of Covid-19, therefore, we were not afforded a space within the hospital premises and had to resort to using the classic method.   

  • Comment: to bring the process closer to a larger scale implementation, it is necessary to introduce some automatisms (perhaps machine learning based on data collected from real users, or programmable software agents) - are you planning any further development to make this method faster and more effective?

Reply: we definitely agree that automating the process could save a lot of time and could help in the process of generalizing the technique to be implemented efficiently, this is still an open field of research that is facing certain challenges especially in agent modeling techniques, while some researchers have previously implemented agent based particle simulations, how closely the behavior of  these agents can resemble the human behavior is still a matter of debate, our group is focused on the development of Isovist Measurements (i.e. area, perimeter,…. Etc.)  based Agent behavior to generalize the process. However, we are still developing our prowess in Automating solutions for such problems.  

We hope that our response was satisfactory and we wish you well

Reviewer 2 Report

The submitted manuscript is more a research experiment report rather than a full-fledged article, although I consider it a solid contribution to the field of user experience research and design. 

My main critique relates to the "results" section. Here, the focus is more on numbers and formulas describing and comparing two experiments, but the qualitative element - the content of what all these numbers arose from - is missing. Where did the new navigation system come from? How was it developed? What does "based on the response of the first group" mean? This is the major shortcoming of this manuscript at the moment. And it is obviously easy to fix: it is clear from the other parts that the authors are fully in control of the research context and of the data and methods. They just need to shift a bit the focus of their narrative from "how/how many times?" to "what is behind those numbers?"

Also, a couple of questions that could, perhaps, guide the revisions:

Did I understand correctly that the purpose of the study was just to test the VRE mechanism, and not to develop a more efficient wayfinding system in the hospital? If the latter, do the authors plan to use qualitative research methods, e.g. participant observation, in-depth interviews, etc., to enrich/supplement quantitative data in the future? To understand what actions, feelings, expectations and behavioral strategies are behind the reduction of time indicators during the transition from the conventional to the refined system of wayfinding.

Author Response

Reply to reviewers’ comments  

  • Comment: My main critique relates to the "results" section. Here, the focus is more on numbers and formulas describing and comparing two experiments, but the qualitative element - the content of what all these numbers arose from - is missing.

Reply: the results section is mainly comprised of the VR experimental results; the nature of our results section was mainly comprised of numerical results and their statistical analysis as the reviewer mentioned in the question. the qualitative aspects of wayfinding in any environment have been mentioned in and discussed rigorously in previous research articles. However, the aim of this research article as mentioned was to examine the effectiveness of the employment of navigable VR environments in the process of optimizing wayfinding in healthcare facilities in Malaysia in general and outpatient units being the focus of this article. Optimizing wayfinding performance in a building generally requires a large number of respondents to prove the effectiveness of the intervention, especially in an outpatient unit where the number of visitors is quite large. Therefore, this study has employed a quantitative research approach, while the qualitative aspects were mentioned in the limitations and future work section of this study to be a limitation to the findings of the study that the research team is aware of.   

  • Comment: Where did the new navigation system come from? How was it developed? What does "based on the response of the first group" mean?

Reply: the new navigation system is based on the original navigation system in the building with added or changed wayfinding devices (primarily in terms of location or orientation of the device), the new additions/ changes are based upon the data collected from users in terms of the Clustering of the pauses in one location which is hypothesized to be a location where there is a lack of provided information, as well as the path tracing collected from the users. A new figure was created and added to the manuscript in attempt to address this question. (Figure. 6)

  • Comment: Did I understand correctly that the purpose of the study was just to test the VRE mechanism, and not to develop a more efficient wayfinding system in the hospital?

Reply: people’s navigation behavior in VR environments is considered by previous researchers to generally resemble actual navigation tasks (From a cognitive perspective), therefore, the assumption is that improving people’s navigation within a virtual environment by implementing a certain logic (in the case of this study: path tracing and locations where users’ suffered a lack of provided information, resulting in some cases to a loss of orientation (The locations where users required The orientation map calls)), would improve users wayfinding in an actual setting, within the limitations of the simulation environment in mind (in this case the Virtual Reality experiment).

  • Comment: do the authors plan to use qualitative research methods, e.g., participant observation, in-depth interviews, etc., to enrich/supplement quantitative data in the future? To understand what actions, feelings, expectations and behavioral strategies are behind the reduction of time indicators during the transition from the conventional to the refined system of wayfinding.

Reply: the authors are considering employing a qualitative method alongside the quantitative method in the future. The results of the study are not without any limitations, the limitations and future works section have encouraged future research initiatives that addresses the qualitative aspects of wayfinding.  

We hope that our response was satisfactory and we wish you well

Reviewer 3 Report

Two virtual reality experiments in this study were carried out, the first representing the existing wayfinding system and the second representing the new distribution. The results indicated an overall reduction of time consumed to reach the same destinations and lesser distances traveled within the environment resulting from the implementation of the new distribution. It is a practical study.

However, the authors claim…this study proposes a way of showing an items distribution scheme to be implemented in healthcare facilities of Malaysia, with outpatient units being the subject of experimentation and the main focus. Whereby a distribution of way showing items based on the outcomes of a virtual reality experiment to evaluate the current wayfinding system. It is difficult to understand how this statement relates to wayfinding.

Moreover, the authors said the institutional guidelines represented by the ministry of health's information management system guidelines give an indication of the expected workflow within each department of the hospital outpatient units included. In fact, there are specific steps in the Wayfinding design, which usually include information search, decision making, and decision execution. Please try to explain how this corresponds to the institutional guidelines.

The authors said that the virtual environment is capable of calculating distances traveled by subjects, the time taken to reach their destination, the number of pauses taken during their journey, and the number of map calls. A set of collected data will be gathered automatically by the VWE application itself, gathering information on subjects', time, distance, a trace of the selected path by the subject, a record of points where the users have used the map option, number of times and locations where users have asked for directions.

I don't think subjects' time, distance, or a trace of the selected path by the subject are enough to represent the performance of wayfinding. A complete wayfinding evaluation also considers user detours, error rates, hesitation time, etc. Such evaluation items will help to define the signage design specifications.

The statistical analysis of this study employed to analyze the collected VRE data indicates a significant improvement in the metrics selected by the study. I suggest that the authors explore the optimal wayfinding design for this site, such as the best setting location, and content.

Author Response

Reply to reviewers’ comments

First and foremost, we would like to thank the reviewer for his valuable comments. 

Reviewer no.3

  • Comment: the authors claim…this study proposes a way of showing an items distribution scheme to be implemented in healthcare facilities of Malaysia, with outpatient units being the subject of experimentation and the main focus. Whereby a distribution of way showing items based on the outcomes of a virtual reality experiment to evaluate the current wayfinding system. It is difficult to understand how this statement relates to wayfinding.

Reply: the premise of this statement was based on the importance of way showing devices in a wayfinding system, whereby it was considered by previous researchers to be an essential part of any wayfinding system.

With regards to the virtual reality experiments, this study has employed this technique to collect data from users to be the basis of the development of the distribution of way showing devices, the assumption here is that users whom may lose their orientation within the building premises (the virtual premises in this instance), are more likely to have encountered a situation where the information provided by the environment wasn’t adequate.  

  • Comment: the authors said the institutional guidelines represented by the ministry of health's information management system guidelines give an indication of the expected workflow within each department of the hospital outpatient units included. In fact, there are specific steps in the Wayfinding design, which usually include information search, decision making, and decision execution. Please try to explain how this corresponds to the institutional guidelines.

Reply: Malaysia is a country that has not yet developed its institutional guidelines with regards to the design and development of wayfinding systems in healthcare facilities, therefore, the institutional guidelines mentioned above were regarding the procedure of handling patients’ referrals and payments (if any). The correspondence is with regards to the referral process which was then used to form the basis for the internal referral system employed in the virtual environment.

  • Comment: The authors said that the virtual environment is capable of calculating distances traveled by subjects, the time taken to reach their destination, the number of pauses taken during their journey, and the number of map calls. A set of collected data will be gathered automatically by the VWE application itself, gathering information on subjects', time, distance, a trace of the selected path by the subject, a record of points where the users have used the map option, number of times and locations where users have asked for directions

I don't think subjects' time, distance, or a trace of the selected path by the subject are enough to represent the performance of wayfinding. A complete wayfinding evaluation also considers user detours, error rates, hesitation time, etc. Such evaluation items will help to define the signage design specifications.

Reply: In addition to the aforementioned measurements, a path tracing technique was employed, which reveals the users’ wayfinding behavior within the environment (see fig. 6).  A deeper and more detailed analysis of users’ wayfinding behavior based on their tracking data is possible, however, it would make the paper way longer and might distract the reader away from the study’s main aim, which is the employment of Virtual reality experiments as a tool to test and develop wayfinding systems at healthcare facilities. The authors plan to further analyze the tracing data, however, it’s a task that would take a long time to perform giving that we have collected data from 260 respondents. Moreover, we added the measurands that the reviewer had suggested in the section of limitations and future works section.     

  • Comment: The statistical analysis of this study employed to analyze the collected VRE data indicates a significant improvement in the metrics selected by the study. I suggest that the authors explore the optimal wayfinding design for this site, such as the best setting location, and content.
  • Reply: The study has employed two VR experiments due to the fact that it is difficult to establish a performance benchmark, that’s why we used a virtual replica of the existing building, and compared the second set of data to indicate an overall improvement.

We hope that our response was satisfactory and we wish you well

Reviewer 4 Report

The strength of the present manuscript lies in the VR methods used to examine the wayfinding issue in the case of hospital. I also appreciate the clarity with which the authors illustrate schematics associated with their approaches and results. Having said this, I do have some comments that I hope the authors could address.

  1. In the section of introduction, the authors mentioned that way showing items are an essential part of any wayfinding system and there are three primary types. It’s really a pity that in the section of results, no discussion was made about the different influences of these different types of way showing items.
  2. The authors may need to reconsider the selection of keywords. Wayfinding should be definitely included. Signage is one of the most important wayfinding elements. More discussion is needed, if the authors include it as a keyword.
  3. The location of way showing items should be specified in the floorplans. Especially because the focus of the discussion section is centered around the comparison between the 1st and 2nd VREs, there should be a figure showing the difference made to improve the existing wayfinding environment.
  4. The resolution of images and maps may need to be improved. For example, the names of maps and spaces are both unreadable in Figure 4.
  5. In page 10, the authors mention the use of M button for displaying map, but in the instruction (Figure 5) this part of explanation is missing.
  6. The readers may have a clearer view of the maps, if Figure 8 and 9 can be merged into one with the locations of “pauses” and “map calls” represented as two different symbols.
  7. The authors have recently published a paper examining the wayfinding issue in the same hospital. If possible, the author may explain the findings of their previous work and the relationship with this manuscript in the introduction.

Author Response

Reply to reviewers’ comments

First and foremost, we would like to thank the reviewer for his valuable comments. 

Reviewer no.4

  • Comment: In the section of introduction, the authors mentioned that way showing items are an essential part of any wayfinding system and there are three primary types. It’s really a pity that in the section of results, no discussion was made about the different influences of these different types of way showing items.

Reply: generally speaking, directional types indicate the direction of a certain space or function within the environment, while identification items identify the space, as for the orientational devices they are devices that aid users’ in knowing their respective location within the environment (such as you are here maps). In this study, no investigation to the effects of an individual type have been conducted, the limitations section mentions the content of way showing devices as a limitation of this study that researchers are encouraged to take on in future research attempts. The reason for this limitation is that by including a map call function, which indicates that users’ have lost their orientation, may have limited the interaction of the subjects with the existing orientation devices due to the fact that subjects were aware of the existence of the map call function. Therefore, as the reviewer may have noticed that the devices that have been added or removed and replaced, were all directional devices. As for the identification devices, no shortage was observed, neither during the site surveys (which were the basis for the distribution during the 1st VRE), nor during the conduct of the 1st VRE which was the basis of the new system. a sentence   

  • Comment: The authors may need to reconsider the selection of keywords. Wayfinding should be definitely included. Signage is one of the most important wayfinding elements. More discussion is needed, if the authors include it as a keyword.

Reply: the keywords were updated to include wayfinding as a keyword, while “signage” was omitted  

  • Comment: The location of way showing items should be specified in the floorplans. Especially because the focus of the discussion section is centered around the comparison between the 1st and 2nd VREs, there should be a figure showing the difference made to improve the existing wayfinding environment.

Reply: a graph was created and included to illustrate of the way showing devices.

  • Comment: The resolution of images and maps may need to be improved. For example, the names of maps and spaces are both unreadable in Figure 4.

Reply: the image was replaced with a new image with a higher resolution.

  • Comment: In page 10, the authors mention the use of M button for displaying map, but in the instruction (Figure 5) this part of explanation is missing.

Reply: the instruction did not mention that on the user interface, however, all the subjects were verbally informed of the option

  • The readers may have a clearer view of the maps, if Figure 8 and 9 can be merged into one with the locations of “pauses” and “map calls” represented as two different symbols.

Reply: A merging of these graphs is possible, however, we chose not to because of the potential overlap between the points (most map call locations if not all are registered as pauses) which would make it difficult to differentiate. Furthermore, if we chose to do that, then the reader may wrongfully understand that the number of pauses was way less than the actual number. 

  • Comment: The authors have recently published a paper examining the wayfinding issue in the same hospital. If possible, the author may explain the findings of their previous work and the relationship with this manuscript in the introduction.

Reply:  the previous work was a survey questionnaire that was conducted at the same facility. The main aim was to establish technique (questionnaire survey) that was built upon the British NHS wayfinding evaluation survey, where the need to upgrade a wayfinding system within the Malaysian public healthcare sector can be established. Our previous research was a motivating factor in initiating this research.

We hope that our response was satisfactory and we wish you well

Round 2

Reviewer 1 Report

Dear authors, thank you for taking my comments into account in the text, and for engaging in discussion in response to the questions raised about the further development of the proposed method.  In my opinion, the article is ready for publication.

Reviewer 4 Report

Thank you for your thoughtful responses to my comments and suggestions. 

This manuscript is a resubmission of an earlier submission. The following is a list of the peer review reports and author responses from that submission.